# Ionic Levothyroxine Formulations: Synthesis, Bioavailability, and Cytotoxicity Studies

**DOI:** 10.3390/ijms24108822

**Published:** 2023-05-16

**Authors:** António Barreira, Andreia F. M. Santos, Madalena Dionísio, Ana R. Jesus, Ana Rita C. Duarte, Željko Petrovski, Ana B. Paninho, Márcia G. Ventura, Luis C. Branco

**Affiliations:** LAQV-REQUIMTE, Department of Chemistry, NOVA School of Science and Technology, NOVA University of Lisbon, Campus de Caparica, 2829-516 Caparica, Portugal; a.barreira@campus.fct.unl.pt (A.B.); afm.santos@campus.fct.unl.pt (A.F.M.S.); madalena.dionisio@fct.unl.pt (M.D.); ar.gameiro@fct.unl.pt (A.R.J.); ard08968@fct.unl.pt (A.R.C.D.); z.petrovski@fct.unl.pt (Ž.P.); inespaninho@fct.unl.pt (A.B.P.)

**Keywords:** levothyroxine (T4), ionic formulations, T4 based ionic liquids (T4-ILs), solubility, permeability, cytotoxicity

## Abstract

Thyroid diseases affect a considerable portion of the population, with hypothyroidism being one of the most commonly reported thyroid diseases. Levothyroxine (T4) is clinically used to treat hypothyroidism and suppress thyroid stimulating hormone secretion in other thyroid diseases. In this work, an attempt to improve T4 solubility is made through the synthesis of ionic liquids (ILs) based on this drug. In this context, [Na][T4] was combined with choline [Ch]^+^ and 1-(2-hydroxyethyl)-3-methylimidazolium [C_2_OHMiM]^+^ cations in order to prepare the desired T4-ILs. All compounds were characterized by NMR, ATR-FTIR, elemental analysis, and DSC, aiming to check their chemical structure, purities, and thermal properties. The serum, water, and PBS solubilities of the T4-ILs were compared to [Na][T4], as well as the permeability assays. It is important to note an improved adsorption capacity, in which no significant cytotoxicity was observed against L929 cells. [C_2_OHMiM][T4] seems to be a good alternative to the commercial levothyroxine sodium salt with promising bioavailability.

## 1. Introduction

Thyroid hormones are essential for the normal growth and energy metabolism of most tissues in the body at all stages. Hypothyroidism is a clinical disorder that occurs when the thyroid gland, located in the neck, does not produce enough thyroid hormone for the body’s requirements. This disease is the second most common endocrine disorder, following diabetes, affecting up to 5% of the general population, with a further estimated 5% being undiagnosed [1]. It is known that the prevalence increases with age and is higher in females than in males [2]. People with this condition may have different changes in their bodies, which can evolve into hypertension, dyslipidemia, infertility, cognitive impairment, and neuromuscular dysfunction [3]. This thyroid dysfunction is diagnosed biochemically as overt primary hypothyroidism, defined as serum thyroid-stimulating hormone (TSH) concentrations above and thyroxine concentrations below the normal reference range [4]. In most patients, alleviation of symptoms can be accomplished with oral administration of synthetic levothyroxine, requiring, in several cases, a lifelong therapy [5]. When full replacement of thyroxine using levothyroxine is needed, the addition of triiodothyronine in a low dosage can be considered for people that continue to have mood or memory issues [6]. However, effective and safe alternatives to the oral administration of levothyroxine should be pursued in the absence of parenteral alternatives [7].

Levothyroxine (T4) can be administered orally as sodium salt, whose bioavailability is quite variable, ranging from 50 to 80% [8]. Solid dosage forms are formulated with pentahydrate, and precision in the amount of drug administered to the patient is required and challenging due to its low dosage (25–300 μg/dose) [9]. Moreover, the absorbed amount, speed of absorption, and bioavailability of the drug are controlled by its solubility and gastrointestinal permeability [10].

Active pharmaceutical ingredients (APIs) are commercialized in crystalline forms. In fact, 40 to 70% of the drugs under development present low water solubility, which might compromise the bioavailability and therapeutic efficacy, originating a possible toxicity increase with side effects [11,12]. There are several methods to improve the solubility, including salt formation. One interesting method is converting APIs into ionic liquid form to avoid polymorphism and greatly enhance their bioavailability, dissolution rates, and stability [13].

Ionic liquids (ILs) are low melting organic salts composed of an organic cation and an organic or inorganic anion, displaying unique features (e.g., higher chemical and thermal stability), being versatile in terms of the chemical structure design [14,15]. Some publications reported the use of ionic liquids to improve the water solubility of drugs, such as diclofenac, ibuprofen, ketoprofen, naproxen, sulfadiazine, sulfamethoxazole, and tolbutamide [16]. It has been proven that combining ibuprofen as an anion with ammonium, imidazolium, or pyridinium cations promotes higher solubility in water and biological fluids, as well as a lower degree of polymorphism and systemic toxicity [17]. Another study using cholinium-based ionic liquids showed an increase of naproxen water solubility up to 600-fold [18]. Moreover, the solubility of the drugs can also be enhanced by using tetrabutylphosphonium to transform acidic APIs (diclofenac, ibuprofen, ketoprofen, naproxen, sulfadiazine, sulfamethoxazole, and tolbutamide) into API-ILs [16,17,18,19]. Ampicillin is also known as a drug associated with low water solubility and consequently low bioavailability [20,21,22]. To date, the authors are not aware of similar works involving the use of T4 or other hormones with ionic liquids to enhance bioavailability. However, there are other approaches to improve solubility and bioavailability. For example, the poorly water solubility of progesterone can be increased by forming inclusion complexes with derivatized cyclodextrins or by the formulation of ternary complexes of progesterone/β-cyclodextrin inclusions with PEG 6000 [23].

The aim of this work is to improve the bioavailability of T4 through the synthesis of API-ILs based on the combination of choline and 1-ethanol-3-methylimidazolium cations with T4 as anion. Complementary studies, including solubility, permeability, and toxicity of T4-based ILs, have been performed. It is known that the amount of API administered for thyroid hormone replacement therapy is usually below 200 µg.

## 2. Results and Discussion

### 2.1. Synthesis and Characterization of API-ILs Based Levothyroxine

#### 2.1.1. Synthesis of T4-ILs

Choline-based ILs have been broadly used and applied in biological and pharmaceutical fields, taking advantage of their non-toxicity and biodegradability [24]. In addition, several works focused on the synthesis and application of choline derivatives for many applications, including biotechnology [25], have been published.

Herein, levothyroxine [T4]^−^ as anion was combined with choline [Ch]^+^ and 1-(2-hydroxyethyl)-3-methylimidazolium [C_2_OHMiM]^+^ cations. The synthetic procedure was optimized, using a simple anionic exchange by metathesis reaction, starting from T4 sodium salt and the correspondent halide cations. Figure 1 illustrates the synthesis of T4-based API-ILs. The desired [Ch][T4] and [C_2_OHMiM][T4] were obtained in moderate to high yields (75% for the former and 81% for the latter) and relatively high purity levels.

#### 2.1.2. Characterization of T4-ILs

The elucidation of the chemical structure of each T4-IL was performed by ^1^H and ^13^C-NMR (Appendix A), ATR-FTIR, and elemental analysis (C, H, and N). ^1^H-NMR of both prepared T4-ILs allowed to identify all expected protons from the cation and anion structures as well as the cation-anion (1:1) proportion. Figure 2 displays the comparative ATR-FTIR spectra of the original T4 sodium salt and each prepared T4-IL.

The T4 sodium salt spectrum presents the OH stretching vibration at 3500–3450 cm^−1^, representing the crystallization of water, sustaining the pentahydrate form in T4. The 3043 and 2940–2868 cm^−1^ signals correspond to stretching vibrations of the C–H from the methylene group and aromatic rings. The band at 1620 cm^−1^ is the bending vibration of the latter bonds, and others identified at 1536, 1500, and 1420 cm^−1^ represent the stretching vibration of the C=C bonds in aromatic systems. At 1560 and 1370 cm^−1^, the symmetric and asymmetric stretching vibrations of the C=O bond from the carboxylate anion are revealed, while, at 1183 cm^−1^, it is possible to see the vibration of the C–O. The bending vibrations of the aromatic C–H are displayed at 1162, 920, and 878 cm^−1^. Finally, the C–N bond is observed at 1053 cm^−1^. These results are compatible with data presented in the literature [26,27]. The bands described above can be found in the spectra of [Ch][T4] (980–900 cm^−1^, belonging to a specific group of quaternary ammonium compounds, is indicative of the presence of the cation) and [C_2_OHMiM][T4] (the signals at 1600 cm^−1^ and 1328 cm^−1^, correspond to the C=C stretching vibration and the imidazolium ring, respectively).

##### Thermal Characterization

Since knowledge of phase transformations and temperature resistance behavior of APIs is a crucial step when developing novel formulations, levothyroxine and both formulations were characterized by Thermogravimetric Analysis (TGA) until 500 °C and by Differential Scanning Calorimetry (DSC) between −90 and 150 °C. The respective thermograms are depicted in Figure 3. Moreover, the starting ionic liquids ([Ch][Cl] and [C_2_OHMiM][Br]) were submitted to a similar DSC procedure to evaluate their thermal events.

Regarding the TGA results (Figure 3a), the weight loss curve found for neat T4 indicates a ~10% weight reduction up to 100 °C due to moisture loss from loosely bound adsorbed and coordinated water, in agreement with the thermal studies already published in the literature [28].

On the other hand, no significant weight loss was registered until 140–150 °C for [Ch][T4] and [C_2_OHMiM][T4], which suggests that levothyroxine in both formulations loses the coordinated water. The respective ~2% mass reduction is associated with adsorbed water evaporation, which is readily regained after the pre-drying treatment. Furthermore, for neat [Na][T4], the onset of the major decrease was detected at 175 °C, following a two-step degradation profile. A similar profile is detected for both formulations, although with an onset slightly shifted to lower temperatures: ΔT_ons_ ~50 °C and ΔT_ons_ ~30 °C, respectively for [Ch][T4] and [C_2_OHMiM][T4].

The calorimetric analysis (Figure 3b-c) seems to confirm the dehydration behavior found by TGA: a multiple endotherm profile, attributed to the removal of adsorbed and coordinated water, is detected from ~35 to ~140 °C for neat T4. The registered thermogram is in accordance with previous studies [28,29], but the temperature location is shifted due to different water contents. A closer inspection of the cryogenic temperature range shows crystal-crystal transitions in T4 (−40 °C on cooling and −34 °C upon heating), coherent with polymorphism. In fact, it is already reported in the literature that [Na][T4] can exist in distinct polymorphic forms [29,30]. It is worthwhile noting that the conversion between polymorphs is reversible, as proved by their detection in the first two cycles carried out up to 40 °C (see the inset of Figure 3b), a temperature insufficient to promote water evaporation. After heating up to 150 °C, water is successfully removed, and no discontinuity in the heat flow curve is observed in the subsequent cooling and heating runs (dashed lines in Figure 3b). Moreover, if the commercial T4 is vacuum dried prior to the calorimetric measurements, no evidence of crystal-crystal transitions is detected (Appendix A).

The combination of levothyroxine with the two organic cations herein studied (Figure 3c) also leads to the disappearance of polymorphism, as only the broad endotherm assigned to water removal was detected in [Ch][T4] and [C_2_OHMiM][T4]. After dehydration, no calorimetric response is registered for both formulations, as previously found for neat T4, indicating that the thermal behavior is dominated by the API, since the starting ionic liquids always display thermal events under thermal cycling (Appendix A). The DSC traces of both formulations only exhibit the broad endotherm compatible with the elimination of freely adsorbed water.

Therefore, TGA and DSC studies allowed to conclude that both formulations promoted the loss of levothyroxine’s coordinated water, also endorsing the disappearance of the polymorphism addressed to [Na][T4]. This is a relevant aspect in the context of the pharmaceutical industry, in which polymorphism could be a major drawback with the manifestation of unwanted effects [31].

### 2.2. Bioavailability Studies

#### 2.2.1. Solubility Assays

Oral levothyroxine is primarily indicated for hypothyroidism therapeutics, but its bioavailability can be restricted by many conditions, such as interfering with medicaments and foods, concomitant diseases, and non-compliance. Several aspects influence oral bioavailability, including drug permeability, first-pass metabolism, and dissolution rate, among others, with aqueous solubility being the most frequent cause [32]. Sodium levothyroxine showed low water solubility, which is confirmed by the literature values (0.150 mg mL^−1^, 25 °C, pH 7.4) [33]. Table 1 presents the obtained solubility data for [Na][T4], [Ch][T4] and [C_2_OHMiM][T4] in different media and temperatures. 

For the synthesized T4-ILs, it is possible to observe an increase in water solubility at 25 °C up to two times more than the original T4, and a significant improvement (up to three times) is also detected at 37 °C. As mentioned before, both organic salts [Ch][Cl] and [C_2_OHMiM][Br] have the ability to enhance low drug solubility. When levothyroxine is added to a phosphate buffered saline (PBS) solution, its solubility is remarkably lower than in water. Evert and co-workers [34] reported that sodium levothyroxine solubility in PBS at 38 °C ranges between 2.5–5.0 (×10^−5^) mol L^−1^. Thus said, solubility assays in PBS at 37 °C were performed with [Na][T4], [Ch][T4] and [C_2_OHMiM][T4], leading to interesting results that, once again, turned our attention to the formulations herein developed. It is possible to observe a boost in solubility up to three times when compared with the original drug. On the other hand, serum solubility assays at 37 °C showed slightly better results for [Na][T4] than for the synthesized T4-ILs.

#### 2.2.2. Permeability Assays

Since permeability and diffusion of [Na][T4] can be modified when combined with ILs ([Ch][Cl] and [C_2_OHMiM][Br]) in aqueous media, solution studies of [Na][T4] and T4-ILs at 37 °C, pH 7.4, were performed for a maximum levothyroxine concentration of 50 mg L^−1^. In this context, a polyethersulphone (PES-U) membrane was selected for the determination of these two parameters. The diffusion coefficient is associated with the amount of API diffused with time. Therefore, the increase in the diffusion coefficient is proportional to the speed of the drug’s diffusion through the membrane. Another aspect to consider is that the solubility of the drug is proportional to the driving force.

Regarding the results obtained for the 8-hour permeability assays (see Table 2), and despite the higher solubility determined in PBS at 37 °C, the values found for permeability and diffusion of the synthesized T4 salts are lower than those calculated for [Na][T4]. The greatest difference is observed for [Ch][T4], with a permeability value of 0.61 cm s^−1^ and diffusion of 0.10 cm^2^ s^−1^. For this salt, only 63% of the mass in the initial solution is identified in the receptor solution at the end of the 8-hour assay. The highest partition coefficient (K_d_) value associated with [Ch][T4] (0.93) is indicative that the salt diffuses rapidly across the membrane, but retention occurs to some extent.

#### 2.2.3. Cytotoxicity Assays

Cell viability tests were performed in L929 cells, commonly used for cytocompatibility studies [35,36]. Figure 4 exhibits the viability of the L929 cells after 24 h exposure to [Na][T4], [Ch][T4] and [C_2_OHMiM][T4] at 50 and 75 ppm. The presented data represent means ± SD (*n* = 3), in which statistically significant differences were determined by Tukey’s multiple comparisons test, a two-way ANOVA.

It is worth noting that L929 cells can endure concentrations between 50 and 75 ppm without losing their viability. Additionally, these results suggested that the two concentrations of the T4-ILs herein explored do not impact the cytotoxicity. In fact, after 24 h of exposure to [Ch][T4] and [C_2_OHMiM][T4], it is possible to assume that T4-ILs are non-toxic to the cells, as no significant differences between the test and the untreated control (DMSO) groups were found.

## 3. Materials and Methods

### 3.1. Reagents

Levothyroxine sodium salt pentahydrate (≥98%, HPLC) was purchased from Sigma-Aldrich (Burlington, Massachusetts, United States). Choline chloride (≥97%, C_5_H_14_ClNO) was acquired on Fluka (Geel, Belgium). Both reagents were used without previous purification. The 1-(2-hydroxyethyl)-3-methylimidazolium bromide ([C_2_OHMiM][Br]) was previously synthesized [37].

### 3.2. Synthesis of API-Ionic Liquids Based on Levothyroxine

#### 3.2.1. Synthesis of Choline Levothyroxine, [Ch][T4]

Choline chloride (14.9 mg, 0.14 mmol, 1.2 equiv.) was dissolved in ethanol in a round-bottomed flask, and then sodium levothyroxine (100 mg, 0.12 mmol, 1 equiv.) was slowly added. The reaction mixture was stirred at room temperature for 24 h. Afterwards, the solution was filtered, and the solvent was evaporated and dried under vacuum. The final product was obtained as a hygroscopic pale-yellow solid (78.3 mg, 75%).

^1^H-NMR (400 MHz, (CD_3_)_2_SO): δ = 7.78 (s, 2H), 6.94 (s, 2H), 3.85 (m, 2H), 3.42–3.40 (m, 2H), 3.12 (s, 9H), 3.08 (m, 1H), 2.80–2.75 (m, 2H) ppm.

^13^C-NMR (100 MHz, (CD_3_)_2_SO): δ = 152.80, 143.93, 141.25, 125,17, 92.54, 88.60, 67.52, 67.49, 67.47, 55.64, 53.69, 53.65, 53.61 ppm.

ATR-FTIR: ν = 3398, 2921, 1603, 1538, 1380, 1223, 1180, 1082, 950, 900 cm^−1^.

Elemental analysis calculated (%) for C_20_H_24_N_2_O_4_I_4_·9H_2_O (1025.62 g.mol^−1^): C 23.40, N 2.73, H 2.34; found: C 23.20, N 2.82, H 2.21.

#### 3.2.2. Synthesis of 1-(2-Hydroxyethyl)-3-Methylimidazolium Levothyroxine, [C_2_OHMiM][T4]

The 1-(2-hydroxyethyl)-3-methylimidazolium bromide (22 mg, 0.17 mmol, 1.5 equiv.) was dissolved in ethanol in a round-bottomed flask, and then sodium levothyroxine (100 mg, 0.12 mmol, 1 equiv.) was slowly added. The reaction mixture was stirred at room temperature for 24 h. Then, the solution was filtered, and the solvent was evaporated and dried under vacuum. The final product was obtained as a hygroscopic pale-yellow solid (88 mg, 81%).

^1^H-NMR (400 MHz, (CD_3_)_2_SO): δ = 9.15 (s, 1H), 7.78 (s, 2H), 7.73(d, 2H, J = 12 Hz), 6.55 (s, 2H), 4.23 (t, 2, H, J = 8 Hz), 3.87 (s, 3H), 3.73 (t, 2H, J = 8 Hz), 2.78 (m, 1H), 2.80–2.75 (m, 2H) ppm.

^13^C-NMR (100 MHz, (CD_3_)_2_SO): δ = 153.03, 141.26, 137.45, 125.21, 123.77, 123.17, 92.53, 88.58, 59.84, 55.86, 52.07, 36.18 ppm.

ATR-FTIR: ν = 3398, 3146, 2851, 1614, 1536, 1398, 1240, 1162, 1053, 830 cm^−1^.

Elemental analysis calculated (%) for C_21_H_21_N_3_O_5_I_4_·8H_2_O (1046.62 g.mol^−1^): C 24.00, N 4.00, H 2.00; found: C 23.97, N 3.59, H 2.16.

### 3.3. Characterization of API-ILs Based on Levothyroxine

#### 3.3.1. Chemical Characterization (NMR, ATR-FTIR, and Elemental Analysis)

^1^H and ^13^C Nuclear Magnetic Resonance (NMR) spectra were recorded on a Bruker AMX400 spectrometer (Zurich, Switzerland), using deuterated DMSO as solvent, and the chemical shifts were reported downfield in parts per million. Attenuated Total Reflectance Fourier Transform Infrared Spectroscopy (ATR-FTIR) spectra were carried out on a PerkinElmer Two infrared spectrometer equipped with a universal attenuated total reflectance sampling accessory. The respective spectra were obtained through Spectrum 10 software, also from PerkinElmer, and the samples were analyzed in the 400–4000 cm^−1^ spectrum range. Elemental analyses (C, N, and H) were performed on an elemental analyzer from Thermo Finnigan-CE Instruments Flash EA 1112 CHNS series (Italy) at the Analysis Laboratory of LAQV-REQUIMTE, Chemistry Department, NOVA School of Science and Technology, Portugal.

#### 3.3.2. Thermal Characterization (TGA and DSC)

Thermogravimetric analyses were performed over a temperature range from room temperature to 500 °C in a thermogravimetric analyzer from Setaram Labsys EVO (France) with a weighing precision of ±0.01%. Each run was conducted at a heating rate of 5 °C min^−1^, under a highly pure argon atmosphere purged at 50 mL min^−1^, with a mass sample between 8 and 10 mg.

Calorimetric studies were carried out on a DSC Q2000 from TA Instruments Inc. (Tzero DSC technology, Guyancourt, France) coupled to an RCS 90 cooling system and operating in the Heat Flow T4P option. Measurements were performed under anhydrous conditions purged with 50 mL min^−1^ of high-purity nitrogen flow. For each experiment, 5–9 mg of sample were weighted and hermetically encapsulated in Tzero aluminum pans. The respective lids were perforated to avoid a pressure increase due to water evaporation. Regarding the experimental procedure, all samples were previously equilibrated at 25 °C and submitted to different thermal treatments. For the samples with levothyroxine, several cooling and heating runs were performed to study the polymorphism addressed to the neat API. In this context, two cycles between −90 °C and 40 °C were conducted at 5 °C min^−1^. Then, the samples were scanned twice at the same rate between −90 and 150 °C. For the starting ionic liquids ([Ch][Cl] and [C_2_OHMiM][Br]), three cycles between −90 and 150 °C at 5 °C min^−1^ were performed. Data treatment was carried out through Universal Analysis 2000 software by TA Instruments Inc.

### 3.4. Solubility Assays

The solubility assays were performed in water (25 °C and 37 °C), phosphate buffered saline (PBS) solution (37 °C), and serum (37 °C) for the synthesized API-ILs and T4. The compounds were weighed, and, with a micropipette, a volume of 1 mL was added multiple times until a homogenous solution was achieved. For the assays at 37 °C, a water bath was prepared, and the same procedure was performed. All the assays were carried out in triplicate. The average and standard deviation were calculated for the three replicates.

### 3.5. Permeability Assays

Permeability measurements were conducted using a glass Franz-type diffusion cell (PermeGear) with an 8 mL reactor compartment and an effective mass transfer area of 1 cm^2^. The chosen membrane was polyethersulphone (PES-U) with 150 µm thickness and 0.45 µm pore size (Sartorius Stedim Biotech, Göttingen, Germany), which was placed between the two compartments and held with a stainless-steel clamp. The receptor compartment was filled with PBS solution, and air bubbles were removed. Afterwards, the donor compartment was loaded with 2 mL of the samples dissolved in ethanol/water (25%/75%) at concentrations of 50 mg L^−1^. Aliquots of 400 µL were withdrawn from the receptor compartment at predetermined time periods (5 min and hourly from 1 to 8 h), and a fresh mixture of ethanol/water (25%/75%) was added to complete the volume. The experiments were performed at 37 °C, and the receptor compartment was stirred at 400 rpm with a magnetic bar to eliminate the boundary layer effect. The determination of the API diffused was performed by HPLC. The assays were performed with three replicates. The average of the relative standard deviation for API diffused amounts was 5%.

The cumulative mass of drug diffused to the receptor compartment was determined taking into consideration the replacement of aliquots with fresh medium and the dilution derived from the addition of fresh buffer. Thus, the permeability, P, of the API through the membrane was calculated according to the equation:(1)$$-ln(1-\frac{2C_{t}}{C_{0}})=\frac{2A}{V}\times P\times t$$
where C_t_ is the concentration in the receptor compartment at time t, C_0_ is the initial concentration in the donor compartment, A is the effective mass transfer area, and V is the total volume of solution in both compartments.

According to a derived solution of Fick’s law of diffusion, it is possible to determine the diffusion coefficient, D, of the API through the membrane. The following equation was, hence, applied to the different systems:(2)$$D=\frac{V_{1}\times V_{2}}{V_{1}+V_{2}}\times \frac{h}{A}\times \frac{1}{t}\ln(\frac{C_{f}-C_{i}}{C_{f}-C_{t}})$$
where C_f_ and C_i_ are the final and initial concentrations in the receptor compartment, and C_t_ is the concentration in the receptor compartment at time t. V_1_ and V_2_ are the volumes in the donor and receptor compartments, respectively, and h is the thickness of the membrane.

### 3.6. Cytotoxicity Assays

#### 3.6.1. Stock solution preparation

Stock solutions of [Na][T4], [Ch][T4], and [C_2_OHMiM][T4] were prepared in DMSO at 10 mg mL^−1^.

#### 3.6.2. Cell Viability Assay

The biological effect of compounds was evaluated on the L929 cell line (DSMZ—German Collection of Microorganisms and Cell Culture GmbH). L929 cells were cultured in Eagle’s Minimum Essential Medium (MEM, with 1.5 g L^−1^ sodium bicarbonate, non-essential amino acids, L-glutamine, and sodium pyruvate, Corning), supplemented with 10% fetal bovine serum (FBS, Corning) and 1% penicillin-streptomycin (Corning). Cells were cultured in a humidified incubator at 37 °C with 5% CO_2_.

Solutions of [Na][T4], [Ch][T4] and [C_2_OHMiM][T4] at 50 and 75 ppm, from a 10 mg mL^−1^ DMSO stock solution, were prepared in MEM and incubated with the cells for 24 h at 37 °C. Cells only incubated with complete media were used as negative control and DMSO was chosen for the untreated cell group. Cell viability was evaluated using the CellTiter 96^®^ Aqueous One Solution Cell Proliferation Assay (Promega), which is based on the tetrazolium active component ((3-(4,5-dimethylthiazol-2-yl)-5-(3-carboxymethoxyphenyl)-2-(4-sulfophenyl)-2H-tetrazolium, MTS). The amount of formazan product was measured in a microplate reader (VICTOR Nivo TM, PerkinElmer, Waltham, MA, USA) at 490 nm, as absorbance is directly proportional to the number of viable cells in culture. Cell viability was expressed as the percentage of cells exposed to extracts vs. controls. Statistical analysis was performed using GraphPad Prism 7.00 software. A two-way ANOVA test was performed as well as Tukey’s multiple comparison test. Statistical differences were considered if *p* < 0.05.

## 4. Conclusions

In this work, two T4-based ILs were synthesized by an anionic exchange reaction, using sodium levothyroxine ([Na][T4]) and two halide salts from choline and 1-(2-hydroxyethyl)-3-methylimidazolium cations ([Ch][Cl] and [C_2_OHMiM][Br]). Both T4-ILs were successfully prepared with high yields and characterized by different spectroscopic techniques, such as ^1^H-NMR, ^13^C-NMR, ATR-FTIR, elemental analysis, TGA, and DSC in order to confirm their chemical structure and thermal stability. It is possible to conclude that no thermal transitions associated to the API’s polymorphism was found on the two prepared T4-ILs, being both thermoresistant up to 175 °C. For the solubility assays, T4-ILs revealed a significant improvement, two to three times higher, in water and phosphate buffer at 25 and 37 °C compared to the commercial [Na][T4]. Regarding the diffusion profiles of T4-ILs, superior results were detected for the formulations herein studied. However, permeability tests revealed a better outcome for [Na][T4] where compared to T4-ILs, being significantly higher than [Ch][T4] and slightly higher than [C_2_OHMiM][T4].

Cytotoxicity assays were performed in L929 cells, and after 24 h exposure to [Na][T4], [Ch][T4] and [C_2_OHMiM][T4] at 50 and 75 ppm, the cells can endure concentrations without losing their viability, making possible to assume that T4-ILs are non-toxic to the cells.

In general, [C_2_OHMiM][T4] seems to be a promising alternative to T4 with higher bioavailability as well as no toxicity.

## Figures and Tables

**Figure 1 ijms-24-08822-f001:**
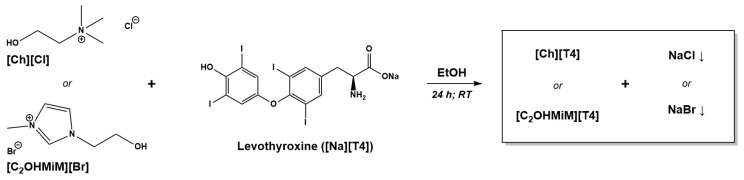
API-IL-based levothyroxine synthesis.

**Figure 2 ijms-24-08822-f002:**
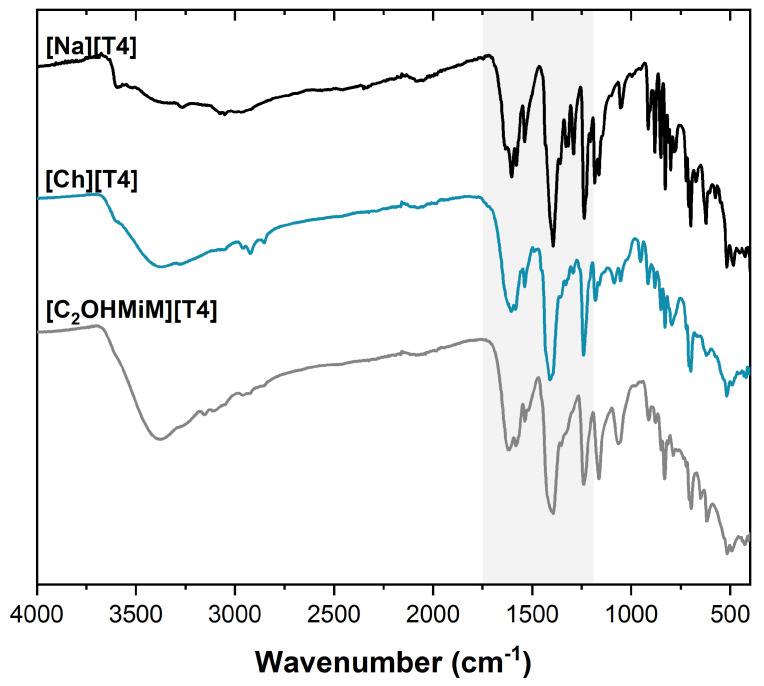
ATR-FTIR comparative spectra for [Na][T4] and T4-ILs.

**Figure 3 ijms-24-08822-f003:**
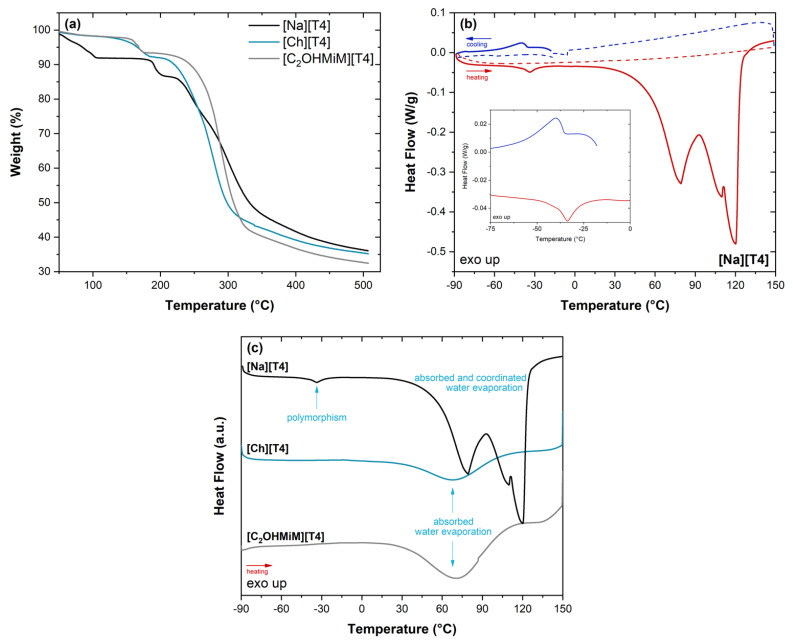
Thermograms for (**a**) the weight loss and heat flow trace of (**b**) neat [Na][T4] and (**c**) comparison between the first heating run of [Na][T4], [Ch][T4] and [C_2_OHMiM][T4].

**Figure 4 ijms-24-08822-f004:**
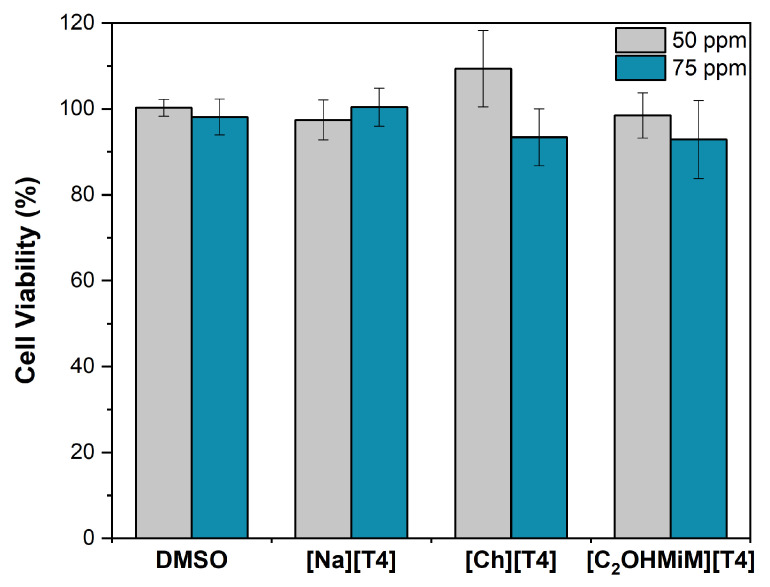
Cell viability towards L929 cells after 24 h exposure to [Na][T4], [Ch][T4] and [C_2_OHMiM][T4] at 50 and 75 ppm. Data illustrate the mean ± SD (*n* = 3), in which statistically significant differences, determined by Tukey’s multiple comparisons test, are represented.

**Table 1 ijms-24-08822-t001:** Water, PBS, and serum solubility data at 25 °C and 37 °C.

Compound	Solubility in Water 25 °C (mg mL^−1^)	Solubility in Water 37 °C (mg mL^−1^)	Solubility in PBS 37 °C (mg mL^−1^)	Solubility in Serum 37 °C (mg mL^−1^)
[Na][T4]	0.149 ± 0.015 ^(1)^	0.155 ± 0.002	0.163 ± 0.004 ^(1)^	0.485 ± 0.005
[Ch][T4]	0.247 ± 0.035	0.386 ± 0.015	0.284 ± 0.017	0.334 ± 0.005
[C_2_OHMiM][T4]	0.277 ± 0.034	0.379 ± 0.022	0.324 ± 0.004	0.321 ± 0.007

^(1)^ [Na][T4] solubility values in water at 25 °C [33] and PBS at 37 °C [34] retrieved from literature.

**Table 2 ijms-24-08822-t002:** Permeability, diffusion, and partition coefficients were calculated.

Compound	Permeability (×10^−5^) cm s^−1^	Diffusion (×10^−6^) cm^2^ s^−1^	K_d_
[Na][T4]	2.04	0.49	0.63
[Ch][T4]	0.61	0.10	0.94
[C_2_OHMiM][T4]	1.02	0.34	0.45

## Data Availability

Data are available in a publicly accessible repository.

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
