# Peer review of "Ionic Levothyroxine Formulations: Synthesis, Bioavailability, and Cytotoxicity Studies"

_ijms, 2023, doi:10.3390/ijms24108822_

Round 1

Reviewer 1 Report

The manuscript report an interesting approach in levothyroxine formulations. It is well written and consider that can be published in its actual form.

Author Response

Reviewer 1 considered that “The manuscript report an interesting approach in levothyroxine formulations. It is well written and consider that can be published in its actual form”.

Authors: The authors appreciate the comment and are thankful to the reviewer.

Reviewer 2 Report

The authors have prepared a research article entitled “Ionic Levothyroxine Formulations: Synthesis, Bioavailability and Cytotoxicity studies”.

Thyroid diseases affect a significant portion of the population, with hypothyroidism being a commonly reported thyroid disease. Levothyroxine (T4) is used clinically to treat hypothyroidism and suppress thyroid-stimulating hormone secretion in other thyroid diseases. However, the narrow therapeutic index of this drug, the need for frequent administration, and the impact of gastrointestinal diseases, foods, and other drugs on its absorption are disadvantages associated with its oral administration. To improve T4 solubility, various approaches such as particle size reduction, nanosuspension, surfactant use, salt formation, and solid dispersion have been explored to enhance drug solubility and bioavailability. This study focuses on the synthesis of T4 salts based on Ionic Liquids (ILs) to enhance drug solubility. The desired TB-based ILs (TB-ILs) were prepared by combining [Na][T4] with choline [Ch] and 1-(2-hydroxyethyl)-3-methylimidazolium [C2OHMIM] cations. The chemical structure, purities, and thermal properties of all compounds were characterized using various techniques. The TB-ILs' solubilities in serum, water, and PBS were compared to [Na][T4], and permeability assays were conducted. The results indicate improved adsorption capacity and no significant cytotoxicity using L929 cells. [C2OHMIM][T4] appears to be a promising alternative to commercial levothyroxine sodium salt, with good bioavailability.

The article has some interesting results and the authors have made considerable attention to preparing it. However, some issues need to be clarified before further consideration. Thus, the reviewer recommends this work can be published after a minor revision.

1.     The abstract is tediously long and it should be modified. Too much background to the study. Particularly, modify the following text and focus directly on the new finding of the current study.

“Thyroid diseases affect a considerable portion of the population, with hypothyroidism being one of the most commonly reported thyroid diseases. Levothyroxine (T4) is clinically used to treat hypothyroidism and suppress thyroid-stimulating hormone secretion in other thyroid diseases. The narrow therapeutic index of this drug, the need for frequent administration as well as the influence of gastrointestinal diseases, foods, and other drugs on its absorption are the shortcomings related to the oral administration of T4. There are several approaches to enhance the drug solubility and bioavailability, such as particle size reduction, nanosuspension, the use of surfactants, salt formation, and solid dispersion, among others.”

2.     A schematic cartoon should be reported to understand the main content of the manuscript

3.     There are plenty of language errors throughout the manuscript, they must be taken care of.  

4.     The quality of Figures must be improved.

5.     The statistical section should be reported in the experimental.

6.     The introduction section is tediously long and should be revised entirely so that the reader can clearly identify the scientific problems solved by this research. The information on biomaterials should be elaborated in the introduction with recent references (preferably 2020-2023). For instance, the author may cover the biomaterials reported by EV Barrera, Jesús Fernando Flores Otero, Ramiro, and Narsimha Mamidi. It would be more realistic to cover such kind of research work in the current manuscript. Which will enrich the quality of the current manuscript as well as the inquisitiveness of the readers.

7.     According to the revised data, the conclusions should be modified with more quantitative data.

Author Response

Reviewer 2 (R2) considered that “The authors have prepared a research article entitled “Ionic Levothyroxine Formulations: Synthesis, Bioavailability and Cytotoxicity studies. The article has some interesting results and the authors have made considerable attention to preparing it. However, some issues need to be clarified before further consideration. Thus, the reviewer recommends this work can be published after a minor revision.”

Authors: We appreciated the comment and are thankful to the reviewer.

R2: The abstract is tediously long and it should be modified. Too much background to the study. Particularly, modify the following text and focus directly on the new finding of the current study.

“Thyroid diseases affect a considerable portion of the population, with hypothyroidism being one of the most commonly reported thyroid diseases. Levothyroxine (T4) is clinically used to treat hypothyroidism and suppress thyroid-stimulating hormone secretion in other thyroid diseases. The narrow therapeutic index of this drug, the need for frequent administration as well as the influence of gastrointestinal diseases, foods, and other drugs on its absorption are the shortcomings related to the oral administration of T4. There are several approaches to enhance the drug solubility and bioavailability, such as particle size reduction, nanosuspension, the use of surfactants, salt formation, and solid dispersion, among others.”

Authors: The abstract was revised and the introductory paragraphs were removed as indicated by reviewer.

R2:     A schematic cartoon should be reported to understand the main content of the manuscript

Authors: A graphical abstract representing the main idea that is subjacent to the reported experimental work was drawn and it is presented below:

R2:     There are plenty of language errors throughout the manuscript, they must be taken care of. 

Authors: We revised the manuscript in order to eliminate the language errors.

R2: The quality of Figures must be improved.

Authors: The quality of all figures was improved with some additional information whenever it was considered relevant.

A new figure, Figure S6 was added to the ESI file:

Figure S6 - Thermogram of neat levothyroxine, evidencing the effect of removing the adsorbed water by drying under vacuum prior to the calorimetric analyses (solid and dashed lines for the first and second runs, respectively). Dotted lines correspond to the DSC curves of [Na][T4] without any pre-treatment. 

R2:     The statistical section should be reported in the experimental.

Authors: The average and standard deviation were calculated for the three used replicates.

The sentences:

“All the assays were carried out in triplicate. The average and standard deviation were calculated for the three replicates.” (L386)

and

“The assays were performed with three replicates. The average of the relative standard deviation for API diffused amount was 5%.” (L403)

were introduced in sections 3.4 and 3.5, respectively.

R2:  The introduction section is tediously long and should be revised entirely so that the reader can clearly identify the scientific problems solved by this research. The information on biomaterials should be elaborated in the introduction with recent references (preferably 2020-2023). For instance, the author may cover the biomaterials reported by EV Barrera, Jesús Fernando Flores Otero, Ramiro, and Narsimha Mamidi. It would be more realistic to cover such kind of research work in the current manuscript. Which will enrich the quality of the current manuscript as well as the inquisitiveness of the readers.

Authors: The introduction was revised and some parts were removed as suggested by reviewer.

R2:  According to the revised data, the conclusions should be modified with more quantitative data.

Authors: The conclusions were revised including some quantitative data.

Reviewer 3 Report

Dear Author,

I would like to inform you that the article entitled “Ionic Levothyroxine Formulations: Synthesis, Bioavailability and Cytotoxicity studies” has been intensively reviewed and evaluated. Firstly, a severe literature review has been performed to investigate its originality. This research study was compatible with the journal’s aim and scope. However, there are some points that need to be revised. Hereby, I would like to present my comments.

Comment_1: (L21) Please explain the abbreviation “…prepared the desired TB based…”

Comment_2: (L60) Please check your expression “… The amount of absorption, speed absorption and…” (as an example: “speed of absorption”, “rate of absorption” could be more convenient.

Comment_3: (L108-109) “In addition, several works focused on the synthesis and application of choline derivatives for many applications including biotechnology.” Please add at least two citations to support “several works”.

Comment_4: All the figures should be cited in the text. (Figure 1 should be cited in the text.)

Comment_5: (L150) Please explain the Asterix in Figure 2. Also, show the specific peaks in the figure.

Comment_6: (L160-161) Please clarify the sentence, the results were compatible with the literature data or not? (“…These results are according to the literature [25], [26].”)

Comment_7: As a general comment the resolution of the figures was not clear and hard to understand, so add high-resolution graphs (e.g., figure 3).

Comment_8: (L220) “centrepiece” please check for the typos (whole manuscript).

Comment_9: (L243, table 1) Please check your solubility method and find better expressions for the solubility data (if possible, add the exact amounts).

Comment_10: (L265) “Kd” Please explain the abbreviation (probably partition coefficient)

Comment_11: (L274-275) “Cytotoxicity assay was performed in L929 cells, commonly used for cytocompability studies”. Please cite at least two studies.

Comment_12: (L304) Please add the city and country of the supplier. As a general comment please apply this to the whole manuscript.

Comment_13: (L343) Please divide them as a separate subheading and indicate the supplier’s cities and countries.

Comment_14: (L375) Please add at least one citation for your solubility assay. Generally, the shake flask method is used for the determination of the exact solubility amount. Please give the exact solubilities of your active compound.

Comment_15: (L441) Please add a finalization paragraph for the overall summary and generalization of the study's outcomes.

Best regards.

Dear Author,

The language of the manuscript should be carefully checked. If possible, a language editing service could be the best option.

Best regards.

Author Response

Reviewer 3 (R3) considered that “I would like to inform you that the article entitled “Ionic Levothyroxine Formulations: Synthesis, Bioavailability and Cytotoxicity studies” has been intensively reviewed and evaluated. Firstly, a severe literature review has been performed to investigate its originality. This research study was compatible with the journal’s aim and scope. However, there are some points that need to be revised. Hereby, I would like to present my comments”.

Authors: The authors appreciate the comment and are thankful to the reviewer.

R3: (L21) Please explain the abbreviation “…prepared the desired TB based…”

Authors: The abbreviation TB as no meaning in the context of this manuscript and it was substituted by T4.

R3:  (L60) Please check your expression “… The amount of absorption, speed absorption and…” (as an example: “speed of absorption”, “rate of absorption” could be more convenient.

Authors: The expression was changed to: “The amount absorbed, speed of absorption and bioavailability of the drug are controlled by the solubility of the drug and its gastrointestinal permeability.”

R3: (L108-109) “In addition, several works focused on the synthesis and application of choline derivatives for many applications including biotechnology.” Please add at least two citations to support “several works”.

Authors: The following references were introduced:

[25]     J. Claus, F. O. Sommer, and U. Kragl, “Ionic liquids in biotechnology and beyond,” Solid State Ion, vol. 314, pp. 119–128, Jan. 2018, doi: 10.1016/j.ssi.2017.11.012.

[26]     S. N. Pedro, C. S. R. Freire, A. J. D. Silvestre, and M. G. Freire, “The role of ionic liquids in the pharmaceutical field: An overview of relevant applications,” International Journal of Molecular Sciences, vol. 21, no. 21. MDPI AG, pp. 1–50, Nov. 01, 2020. doi: 10.3390/ijms21218298.

R3:  All the figures should be cited in the text. (Figure 1 should be cited in the text.)

Authors: The sentence “The T4 based API-ILs synthesis is represented in Figure 1” was introduced in  L113 in order to cite Figure 1. All the other figures were cited in the text.

R3:  (L150) Please explain the Asterix in Figure 2. Also, show the specific peaks in the figure.

Authors: The asterisks were removed since they were not representative of the description in the text. Changes in the text were made accordingly:

“The T4 sodium salt spectrum presents the stretching vibration 3500-3450 cm-1 representing the crystallization water sustaining the pentahydrate form from T4. The 2940-2868 and 3043 cm-1 correspond to stretching vibration of the C-H from methylene group and aromatic rings. The band at 1620 cm-1 is the bending vibration of the latter bonds and others identified at 1420, 1500 and 1536 cm-1 represent the stretching vibration of the C=C bonds from aromatic systems. At 1560 and 1370 cm-1 the symmetric and asymmetric stretching vibration of C=O bond from the carboxylate anion are revealed and also it is possible to see at 1183 cm-1 the vibration of the C-O. The bending vibrations of the aromatic C-H are revealed at 1162, 920 and 878 cm-1. Finally, C-N bond is observed at 1053 cm-1. These results are compatible with the data presented in the literature [25], [26]. The bands described above can be observed in spectra for the synthesized API-ILs [Ch][T4] and [C2OHMiM][T4]. A peak between 900-980 cm-1 belonging to a specific group of quaternary ammonium compounds is indicative of the presence of choline cation in the spectrum of [Ch][T4]. In the spectra of [C2OHMIM][T4] it is observed a band at 1600 cm-1 corresponding to the C=C stretching vibration and the band for the imidazolium ring at 1328 cm-1 characteristics of the IL [C2OHMIM][Br].”

R3: (L160-161) Please clarify the sentence, the results were compatible with the literature data or not?

Authors: The results are according to the literature [25], [26].

R3: As a general comment, the resolution of the figures was not clear and hard to understand, so add high-resolution graphs (e.g., figure 3).

Authors: The quality of all figures was improved with some additional information whenever it was considered relevant.

A new figure, Figure S6 was added to the ESI file:

Figure S6 - Thermogram of neat levothyroxine, evidencing the effect of removing the adsorbed water by drying under vacuum prior to the calorimetric analyses (solid and dashed lines for the first and second runs, respectively). Dotted lines correspond to the DSC curves of [Na][T4] without any pre-treatment. 

R3: (L220) “centrepiece” please check for the typos (whole manuscript).

Authors: We have changed to: “Oral levothyroxine  is primarily indicated for hypothyroidism therapeutics, but its bioavailability can be restricted by many conditions, like interfering with medicaments and foods, concomitant diseases and noncompliance and that is why oral ingestion is the most suitable and commonly used route of drug delivery due to its ease administration.”

R3: (L243, table 1) Please check your solubility method and find better expressions for the solubility data (if possible, add the exact amounts).

Authors: The exact amounts were added to the table 1 and a description for the solubility method was also introduced on the experimental section.

Table 1 - Water, PBS and serum solubility data at 25 °C and 37 °C

Compound

Solubility in water 37 °C (mg/mL)

Solubility in water 25 °C (mg/mL)

Solubility in PBS 37 °C (mg/mL)

Solubility in Serum 37 °C (mg/mL)

[Na][T4]

0,155 ± 0,002

0,149 ± 0,015 (1)

0,163 ± 0,004(1)

0,485 ± 0,005

[Ch][T4]

0,386 ± 0,015

0,247 ± 0,035

0,284 ± 0,017

0,334 ± 0,005

[C2OHMIM][T4]

0,379 ± 0,022

0,277 ± 0,034

0,324 ± 0,004

0,321 ± 0,007

  • [Na][T4] solubility in PBS at 37ºC [35] and [Na][T4] solubility in water 25 ºC [34]

R3:  (L265) “Kd” Please explain the abbreviation (probably partition coefficient)

Authors: The sentence was modified to: “The highest partition coefficient (Kd) value associated with [Ch][T4], 0.93, is indicative that the salt diffuses rapidly across the membrane but retention occurs to some extent.”

R3: (L274-275) “Cytotoxicity assay was performed in L929 cells, commonly used for cytocompability studies”. Please cite at least two studies.

Authors: The following references were introduced:

[36]     D. Campoccia et al., “Exploring the anticancer effects of standardized extracts of poplar-type propolis: In vitro cytotoxicity toward cancer and normal cell lines,” Biomedicine and Pharmacotherapy, vol. 141, Sep. 2021, doi: 10.1016/j.biopha.2021.111895.

[37]     V. Cannella et al., “Cytotoxicity Evaluation of Endodontic Pins on L929 Cell Line,” Biomed Res Int, vol. 2019, 2019, doi: 10.1155/2019/3469525.

R3: (L304) Please add the city and country of the supplier. As a general comment please apply this to the whole manuscript.

R3: (L343) Please divide them as a separate subheading and indicate the supplier’s cities and countries.

The city and country for the supplier was introduced whenever it was possible.

R3: (L375) Please add at least one citation for your solubility assay. Generally, the shake flask method is used for the determination of the exact solubility amount. Please give the exact solubilities of your active compound.

Authors: The solubility assay was determined by visual analysis and the shake flask method was not used. Despite this method the solubility studies were repeated at least three times.

R3: (L441) Please add a finalization paragraph for the overall summary and generalization of the study's outcomes.

Authors: The reviewer is asking for a finalization paragraph summarizing the study’s outcome in L441, located at the end of the experimental description. The authors have not understood the meaning of such a paragraph in this particular section and, for this reason they have choose not to introduce it.
